# Analysis of Intraoperative Variables Responsible for the Increase in Lactic Acid in Patients Undergoing Debulking Surgery

**DOI:** 10.3390/jpm13111540

**Published:** 2023-10-26

**Authors:** Marta Soriano Hervás, Daniel Robles-Hernández, Anna Serra, Rosa Játiva-Porcar, Luis Gómez Quiles, Karina Maiocchi, Sara Llorca, María Teresa Climent, Antoni Llueca

**Affiliations:** 1Department of Anaesthesiology, University General Hospital of Castellon, 12004 Castellon, Spain; rjativa@gmail.com; 2MUAPOS (Multidisciplinary Unit of Abdomino-Pelvic Oncology Surgery), University General Hospital of Castellon, 12004 Castellon, Spain; serraa@uji.es (A.S.);; 3Department of Anaesthesiology, University La Plana Hospital, Road from Vila-Real to Burriana, km 0.5, 12540 Castellón, Spain; 4Department of Obstetrics and Gynaecology, University General Hospital of Castellon, 12004 Castellon, Spain; 5Department of General Surgery, University General Hospital of Castellon, 12004 Castellon, Spain

**Keywords:** cytoreductive surgery, lactic levels, intraoperative peritoneal cancer index, duration of surgery, preoperative haemoglobin, intraoperative bleeding, fluid therapy administered, administration of blood products

## Abstract

**Background**: Cytoreductive surgery (CRS) is a complex procedure with a high incidence of perioperative complications. Elevated lactacidaemia levels have been associated with complications and perioperative morbidity and mortality. This study aims to analyse the intraoperative variables of patients undergoing CRS and their relationship with lactacidaemia levels. **Methods**: This retrospective, observational study included 51 patients with peritoneal carcinomatosis who underwent CRS between 2014 and 2016 at the Abdomino-Pelvic Oncological Surgery Reference Unit (URCOAP) of the General University Hospital of Castellón (HGUCS). The main variable of interest was the level of lactic acid at the end of surgery. Intraoperative variables, including preoperative haemoglobin, duration of surgery, intraoperative bleeding, fluid therapy administered, administration of blood products, and intraoperative peritoneal cancer index (PCI), were analysed. **Results:** Positive correlations were found between lactic acid levels and PCI, duration of intervention, fluid therapy, intraoperative bleeding, and transfusion of blood products. Additionally, a negative correlation was observed between haemoglobin levels and lactic acid levels. Notably, the strongest correlations were found with operative PCI (ρ = 0.532; *p*-value < 0.001) and duration of surgery (ρ = 0.518; *p*-value < 0.001). **Conclusions**: PCI and duration of surgery are decisive variables in determining the prognosis of patients undergoing debulking surgery. This study suggests that, for each minute of surgery, lactic acid levels increase by 0.005 mmol/L, and for each unit increase in PCI, lactic acid levels increase by 0.060 mmol/L.

## 1. Introduction

Peritoneal carcinomatosis (PC) [1] is a clinical entity in which a malignant tumour originating from the peritoneal surface or visceral organs spreads widely throughout the inner surface of the abdomen. This entity has historically been considered an incurable pathology and treated with an abstentionist attitude since its description [2,3].

Sugarbaker [3] found that some patients with PC could benefit from surgical removal of any gross tumour, with the possibility of combining it with intraoperative locoregional chemotherapy. This technique, known as debulking or cytoreductive surgery (CRS), with or without hyperthermic intraoperative chemotherapy (HIPEC), has been used to treat patients with PC of different cell lines (gastric, colorectal, ovarian cancer, etc.) [4]. This strategy has been reported to significantly improve the survival rate of patients with this condition [5]. The use of HIPEC after the completion of extensive peritonectomy procedures has generated over 60% 10- and 15-year survival rates in patients with PC [5]. Previously, survival rates of 5.2 months were described in patients affected by this disease [6].

The metastatic peritoneal involvement that defines PC occurs at the time of diagnosis in up to 7–10% of cases in non-gynaecological neoplasms (gastric, colorectal, pseudomyxoma peritonei, etc.) [6] and in up to 60–70% of patients with ovarian carcinoma [7]. The strongest prognostic factor in patients with advanced ovarian cancer (AOC) is surgery with no residual tumour, and optimal debulking with no macroscopic evidence of residual tumour is associated with a better overall survival rate in the event of disease spreading throughout the peritoneum [8,9]. In this context, an appreciation of the degree of peritoneal carcinomatosis is crucial, given that a description and quantification of the tumour burden are essential for planning surgical procedures, with direct impacts on morbidity and mortality [10].

Thus, the quantification of the PC degree is a prognostic factor [11]. Among the main requirements of a quantification method should be its reproducibility in a defined patient population and its ease of use. Four systems for quantifying the extent of PC have been proposed: the peritoneal cancer index (PCI) [12], the Lyon classification system [13], the simplified classification of the Dutch group peritoneal carcinomatosis index (SPCI) [14], and the Japanese Society for Carcinomatosis in Gastric Cancer Research Classification (JRSGS) [15].

The Lyon classification system was first described in 1994 by Gilly [13]. It considers the size of tumour implants (<5 mm, from 5 mm to 2 cm and >2 cm) and their distribution (localized or diffuse), dividing peritoneal extension into four stages (Table 1).

The Dutch Cancer Institute [14] classifies peritoneal implants, based on their size, into large (>5 cm), medium (1–5 cm), or small (<1 cm). Tumour distribution, in its classification, is recorded according to the presence of tumour deposits in seven areas:-Right subdiaphragmatic region.-Left subdiaphragmatic region.-Subhepatic region.-Omentum and transverse colon.-Small intestine, including its mesentery.-Ileocecal region.-Pelvic region.

In Japan, CP classification of gastric cancer is carried out as follows, according to the JRSGC [15]:-P0: No peritoneal implants.-P1: Peritoneal implants directly adjacent to the stomach’s peritoneum, including major omentum.-P2: Small scattered implants or ovarian metastases.-P3: Numerous diffuse peritoneal implants.

This classification has been used in Japanese studies as a precise quantitative prognostic indicator.

Sugarbaker’s PCI is arguably the most widely used today. Initially proposed by Jacquet and Sugarbaker [12], it involves a quantitative evaluation of the degree of peritoneal extension of the disease. In its calculation, two variables are considered: tumour distribution in the different abdominopelvic regions and size of the lesions. To calculate PCI, the peritoneal cavity is divided into 13 anatomical regions (Figure 1). The size of the lesion considers the maximum diameter of an implant. If a region is affected by numerous implants, the diameter of the largest one is measured and recorded. Primary tumours or recurrences located in the surgical site are excluded from this assessment. An implant’s size allows the tumour area to be classified into four types (LS-0 to LS-3). LS-0 means there are no implants. LS-1 refers to visible implants smaller than 0.5 cm in diameter. LS-2 identifies nodules larger than 0.5 cm and smaller than 5 cm. LS-3 refers to implants greater than 5 cm or to the presence of multiple confluent implants.

The PCI was first described by Sugarbaker in 1998 and was considered as the standard for describing carcinomatosis of colorectal cancer and mesothelioma [12]. Llueca et al. proved the impact and prognostic value of PCI in patients with advanced ovarian cancer (AOC) [16]. Elias et al. [17] reported that survival at 2 and 5 years was significantly higher when the PCI was less than 16 in patients operated on for PC of colorectal origin. Sugarbaker [18] reported that in a series of 100 patients with colon cancer PC, a 5-year survival rate of 50% was observed when the PCI was less than 10, 20% with a PCI between 11 and 20, and 0% if the PCI was greater than 20. In our centre, the General Hospital of Castellón, Llueca et al. [19] have reported a 3-year disease-free survival rate in advanced stages of ovarian cancer. They described PCI as an effective method to quantify tumour burden and demonstrated its prognostic value.

Hence, CRS, associated or not with HIPEC, has been developed over recent decades as a therapeutic option for selected patients with PC. The evidence supports the use of CRS to eliminate all gross disease, associated or not with HIPEC, to eradicate microscopic disease and reduce peritoneal recurrence to improve patients’ prognoses [5].

The performance of this type of surgery is an extensive and complex procedure with a high incidence of perioperative complications [5,20,21]. Complications can be classified according to failure-to-rescue analysis [20] (Table 2) and their severity can be stratified according to the Clavien–Dindo Classification (Table 3) [22,23]. Close monitoring [4,15], performing these procedures in centres with experience in large oncological surgeries [24,25], and appropriate selection of patients [26] can reduce this incidence.

In this context of complex or prolonged surgeries, cellular metabolic requirements increase. This situation can lead to an imbalance in oxygen supply and demand. This imbalance results in a state of tissue dysoxia [27] which is associated with an increase in lactate levels. It has been shown that high levels of lactacidaemia predict a higher incidence of complications [21,28,29] and perioperative morbidity and mortality [18,19]. Therefore, the search for factors that may contribute to the increase in blood lactacidaemia levels is a major challenge in our work. The objective of this study is to analyse various intraoperative variables of patients undergoing CRS and their relationship with lactacidaemia levels.

## 2. Materials and Methods

This was a retrospective, single-centre, observational, clinical study conducted in the Reference Unit Abdomino-Pelvic Oncological Surgery (RUAPOS) of General University Hospital of Castellon (GUHC). Eligible patients were adults with PC who underwent CRS between 2014 and 2016. A total of 51 patients from 57 have been included.

The main variable was the level of lactic acid at the end of surgery (mmol/L). Preoperative haemoglobin (g/dL), duration of surgery (minutes), intraoperative bleeding (mL), fluid therapy administered in the form of crystalloids and colloids (mL), administration of blood products during surgery (mL), and intraoperative PCI have been included as intraoperative variables. The absence of this information was a criterion for exclusion from this study.

All procedures were performed by the same surgical team. All patients were admitted during the immediate postoperative period to the Intensive Care Unit (ICU) of our hospital.

This study was approved by the Ethics and Clinical Research Committee of the General University Hospital of Castellon. All procedures performed in this study were in accordance with the ethical standards of our institutions and in accordance with the Declaration of Helsinki.

The sample size required to meet this study’s objective was calculated based on Hulley et al.’s [30] formula for correlation coefficient:$$n\;={\left.\left(\frac{z_{1-\alpha /2}+z_{1-\beta }}{\frac{1}{2}\ln\left(\frac{1+\rho }{1-\rho }\right)}\right.\right)}^{2}+3$$

An alpha risk (probability that null hypothesis will be rejected when it is true; it is also known as a type I error or a false positive) of 0.05, a beta risk (probability that a false hypothesis is accepted as true; it is also known as a type II error or a false negative) of 0.20, and estimated correlation coefficient of 0.4 were assumed. Considering a bilateral contrast, a sample size of 47 patients was estimated.

## 3. Data Collection

Data on lactic acid levels, measured using arterial blood samples, were collected at the time of admission and discharge from the ICU. Sociodemographic, anaesthesia, and perioperative data were collected from patient medical records.

## 4. Statistical Analysis

For descriptive analysis, qualitative variables are reported with relative and absolute frequencies, while quantitative variables are presented using centrality and dispersion measures (mean and standard deviation [SD]).

To evaluate the correlation between lactic acid levels after surgery and the rest of the intraoperative variables, a bivariate analysis was performed by calculating Pearson’s correlation coefficient. Spearman’s correlation coefficient was applied in the absence of normality; normality was assessed using the Shapiro–Wilk test.

A strong positive (negative) relationship was estimated when the correlation coefficient was close to 1 (−1), while absence of relationship was estimated when the correlation coefficient was close to 0. Results with a *p*-value less than 0.05 were considered statistically significant.

Subsequently, a multivariate analysis was performed with the level of lactic acid as the dependent variable. As independent variables, the variables of the previous bivariate analysis were included; they showed a statistically significant relationship with respect to the level of lactic acid. A multivariate lineal regression model was applied with the *stepwise* independent variable selection method. This method performs iterations by removing and adding the independent variables, thus obtaining a final model with the set of variables that offer a better fit.

Additionally, the correlation between all intraoperative variables was evaluated. To accomplish this, the correlation matrix was calculated using Spearman’s coefficients.

Statistical analysis was carried out using Statistical Package for the Social Sciences (version 24) (IBM Corporation) and R Statistical Software (version 4.0.3; R Foundation for Statistical Computing, Vienna, Austria).

## 5. Results

A total of 51 cases over a two-year period (2014–2016) were included, corresponding to the initial series of patients diagnosed with PC who underwent CRS at the General University Hospital of Castellon. The mean age was 59.86 years (SD: 12.45). Patients were stratified using the American Society of Anesthesiologists’ (ASA) physical status (PS) classification system. In total, 57% of patients were in ASA II (n = 29/51). None of the patients were classified as IV or V. Recurrent tumours were treated in 18 of the 51 patients (35.29%). Less than 25% of patients had a metastatic status (22.45%) and almost 40% received neoadjuvant treatment (n = 19/49). The sociodemographic and clinical variables are shown in Table 4.

The mean values of preoperative haemoglobin and intraoperative PCI were 11.2 g/dL (SD: 1.5) and 14.75 (SD: 9.25), respectively. During the surgeries, there was an average of 2149 mL of blood loss and an average of 8177 mL of fluid therapy was administered (887 mL and 7290 mL of colloids and crystalloids, respectively). The mean duration of surgery was 8 h and 51 min; the mean postoperative lactic acid was 2.6 mmol/L (SD: 1.83). Table 5 shows the operative variables in greater detail.

To analyse the relationship of operative variables with lactic acid levels, Spearman’s correlation coefficients were estimated, since no normality was found in the sample. The strongest correlations were found with operative PCI (ρ = 0.532; *p*-value < 0.001) and with duration of surgery (ρ = 0.518; *p*-value < 0.001). Statistically positive correlations were found for all other variables except preoperative haemoglobin, for which a negative correlation was calculated (ρ = −0.144; *p*-value = 0.313). Table 6 shows all correlation coefficients with their corresponding significance level (*p* value).

A stepwise multivariate linear regression model was used to evaluate the prediction of lactic acid by a subset of these variables. To start the model, the variables whose correlation was found to be statistically significant were chosen (all except haemoglobin).

This resulted in a model with lactic acid as the dependent variable and PCI and duration of surgery as independent variables. The details of the model and goodness of fit are described in Table 7.

Consequently, lactic acid level could be estimated as follows:Lactic=−1.058+0.005∗Duration+0.060∗PCI

In this way, for each minute of surgery, lactic acid would increase by 0.005 mmol/L and for each unit of PCI, lactic acid would increase by 0.060 mmol/L. These results are in line with the estimated correlations, as they are the two variables with the strongest correlations found.

Additionally, the correlation between all clinical operative variables was estimated (Figure 2). The correlation matrix represented by blue circles (positive correlations) and red circles (negative correlations) is shown. Both the intensity of the colour and the size of the circles indicate the strength of the correlation, with intense colour and large circles being the strongest correlations, whether positive or negative.

The strongest correlation was found between fluid therapy and crystalloid volume (ρ = 0.98); this relationship was expected since fluid therapy consisted mainly of crystalloids. A strong positive correlation was also found between intraoperative bleeding and blood products administered (ρ = 0.62), followed by the correlation between lactic acid and PCI (ρ = 0.53).

## 6. Discussion

As previously described [21], lactic acid values were correlated with morbidity and mortality, which are usually considered major outcome variables in CRS. The analyses of data from our study indicate that there is a positive correlation between PCI, duration of intervention, fluid therapy, intraoperative bleeding, and transfusion of blood products with lactic acid levels, as well as a negative correlation between haemoglobin and lactic acid levels. Particularly, the strongest correlations were found with operative PCI and with the duration of surgery.

Multiple studies have evaluated the clinical use of lactate as a predictor of morbidity and mortality in critical [24] and postoperative [21,28,29,31] patients.

Under physiological conditions, various organs (muscles, intestines, red blood cells, the brain, and skin) produce around 1500 mmol of lactate daily [32]. Lactate is metabolized by the liver (60%), kidneys (30%), and other organs [32]. The normal blood lactate concentration is about 1 mEq/L [33]. It is known that an increase in lactic acid is one of the clinical parameters that provides an early indication of the onset of anaerobic metabolism or metabolic imbalance between oxygen supply and demand [34,35]. Small increases in lactate concentrations are associated with a higher mortality rate [33,36].

This study suggests that the intraoperative variables analysed (PCI, bleeding, transfusion of blood products, fluid therapy administered, and duration of surgery), except preoperative haemoglobin, are significantly correlated with lactate levels, specifically highlighting PCI and duration of surgery. The findings of this study suggest that for each minute of surgery, lactic acid would increase by 0.005 mmol/L and for each unit of PCI, lactic acid would increase by 0.060 mmol/L.

In a study we published in 2021, the relationship between lactacidaemia and postoperative complications after CRS was analysed. It was observed that lactic acid levels on admission to the critical care unit were significantly higher in those who suffered complications, with a relative risk of almost threefold (2.9; CI95%: 1.6–5.3) with respect to uncomplicated patients, with 2.5 mmol/L being the cut-off point for an increased risk of complications and death. That means serum lactate level is a predictive factor for postoperative complications in patients undergoing CRS for peritoneal carcinomatosis. The study suggests that the risk of developing severe complications almost triples with a lactic acid level of 2.5 mmol/L or higher at the time of admission to the ICU.

The relationship between PCI and postoperative complications has been analysed in multiple studies, with the cut-off point of PCI varying by publication [26,37,38,39,40]. The work of Chua et al. [39] in patients with PC due to CRS, peritoneal pseudomyxoma, and peritoneal mesothelioma, established a PCI of 17 as a cut-off point from which the incidence of complications rose considerably. Llueca et al. [16] found that PCI >10 was associated with a poor prognosis in patients with advanced ovarian cancer. No evidence of a relationship between PCI and lactic acid had been found prior to this study.

The literature is variable on the duration of the surgical procedure from which an increase in postoperative risk is specified. It depends on the experience of the surgical team, the type of patients treated and the pathology that causes PC. Roviello et al. [40], when referring to ovarian PC treated with debulking and HIPEC, reported an increase in the rate of complications in patients with interventions longer than 8 h. Younan et al. [41] related an operative time greater than 8.7 h with the presence of an increased risk of postoperative digestive complications (perforation, fistula, and anastomotic dehiscence). Our study is in line with these findings. It has been observed that the longer the intervention lasts, the higher the lactic levels of the patient.

As described above, due to its complexity, the surgery is often prolonged. This results in large-volume blood loss and requires significant use of blood product support. Di Giorgio et al. [42] described 84.4% of their operations as taking 6–12 h and 8.8% as more than 12 h in duration. They estimated that the mean blood loss was 1490 mL (range 100–4900 mL) for patients undergoing CRS with HIPEC, which required correction with a mean of 3.4 units of blood and 5.2 units of plasma. Studies at other institutions have suggested as many as 76% of patients having CRS require blood replacement, with almost half (46%) requiring or receiving fresh frozen plasma (FFP) [43]. Adverse effects of allogenic blood transfusions, such as postoperative infection, cancer recurrence, pulmonary function, length of stay, and mortality, have been shown in multiple trials [44,45]. However, no data have been found in the relationship between blood transfusion and lactate. Our paper shows a statistically positive correlation between the levels of lactic acid and the administration of blood products (BP). Obviously, the administration of BP is related to the length of surgery, which in turn is usually related to blood loss or vice versa.

Our data showed that increased haemoglobin is related to lower lactate levels, although this relationship was not statistically significant in our population. Anaemia is increasingly recognized as an interventional haematological target in patients before major surgery. Preoperative anaemia increases the need for perioperative blood transfusion, and there is now a well-recognized association with increased patient complications, length of hospital stays, and worse outcomes [46]. Multiple studies have shown that preoperative anaemia is associated with worse outcomes after cardiac and noncardiac surgeries, with an increase in the number of transfusions, which, in turn, is associated with increased complications [46,47,48,49]. Thus, while our data may not demonstrate a statistically significant relationship between haemoglobin levels and lactic levels, an association was indeed observed. It is important to note that our data were from a small sample of patients which may not have been large enough to detect a significant effect. Therefore, further studies with a larger population should be conducted to make final conclusions.

Regarding fluid therapy, our data coincide with other studies regarding the use of balanced crystalloids as the first fluid option [50]. Perioperative fluid management is a challenge during major surgery. For a long time, fluid administration has been based on empirical values [51]. Large amounts of crystalloids have been administered to replenish deficits caused by fasting, blood loss, diuresis, insensible losses, and the so-called third space. In systematic reviews that mediate extracellular volume, it was concluded that the third space was erroneous, and this concept has now been abandoned. Goal-directed therapy, or zero-balance therapies, have been associated with better surgical outcomes [52,53,54]. In our institution, the trend is to perform goal-guided therapy through minimally invasive monitoring (proAQT). ProAQT is a minimally invasive haemodynamic monitor which enables a reliable and physiological interpretation of the patient’s haemodynamic status [55]. Our data show that greater fluid administration is related to longer surgical times and more complex surgeries, with greater bleeding and higher lactic levels.

One of the strengths of the present study is the homogeneous cohort of patients with abdominal carcinomatosis that has been treated by the same multidisciplinary, specialized team in the context of a tertiary hospital. Further, to our knowledge, this is the first time an association has been found between the variables described and lactate levels. This is of great interest given the known association between lactic acid levels and perioperative complications.

Regarding the limitations of this study, it should be noted that it is an observational study. The data come from a small series and a single centre, so the results should not be generalized. Thus, although our data indicate that there is a positive correlation between PCI, duration of intervention, fluid therapy, intraoperative bleeding, and transfusion of blood products with lactic acid levels, as well as a negative correlation between haemoglobin and lactic levels, the size of this study makes generalizations impossible. Moreover, this study focuses on specific intraoperative variables; however, other potential preoperative factors influencing lactic acid levels, such as a patient’s age, nutritional status, obesity, diabetes mellitus, and tobacco use are not discussed in the article. They can be studied in subsequent investigations.

In the future, deepening these relationships would be an interesting way to improve the perioperative morbidity and mortality of these patients. Developing more studies with a larger number of patients that confirm these relationships, as well as investigating haemodynamic changes through minimally invasive haemodynamic monitors, which also can affect a patient’s prognosis [56], can be of important clinical interest in future research.

## Figures and Tables

**Figure 1 jpm-13-01540-f001:**
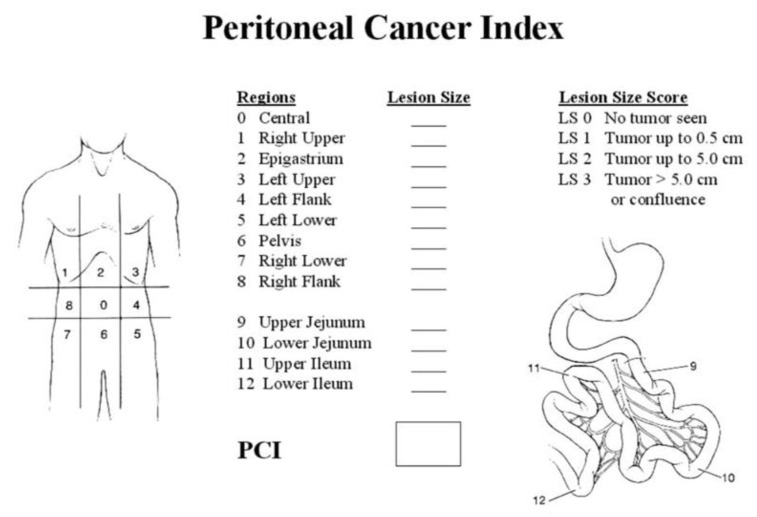
Graphical representation of the Sugarbaker PC index.

**Figure 2 jpm-13-01540-f002:**
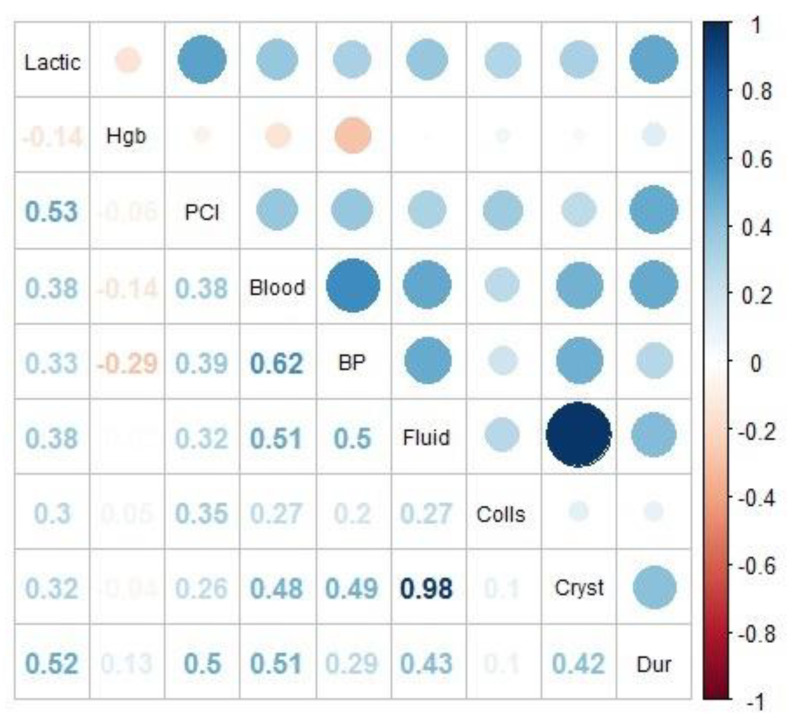
Correlation matrix.

**Table 1 jpm-13-01540-t001:** The Lyon system for classifying peritoneal extension.

Stage	Description of Peritoneal Disease
0	No macroscopic disease
1	Disease <0.5 mm in an abdominal region
2	Disease <0.5 mm with diffuse distribution in abdomen
3	Disease between 0.5 and 2 cm
4	Disease >2 cm

**Table 2 jpm-13-01540-t002:** Complications used in traditional failure-to-rescue analysis. From Silber et al. [20].

**Cardiac**	Arrythmias, arrest, infarction, congestive heart failure
**Respiratory**	Pneumonia, pneumothorax, bronchospasm, respiratory compromise, aspiration pneumonia
**Circulatory**	Hypotension, shock, hypovolaemia
**Neurologic**	Stroke, transient ischaemic attack, seizure, psychosis, coma
**Vascular**	Deep vein thrombosis, pulmonary embolus, arterial clot, phlebitis
**General**	Internal organ damage, return to surgery
**Infection**	Deep wound infection, sepsis, gangrene, amputation
**Others**	Gastrointestinal bleeding, blood loss, peritonitis, intestinal obstruction, renal dysfunction, hepatitis, pancreatitis, decubitus ulcers, orthopaedic complication, compartment syndromes

**Table 3 jpm-13-01540-t003:** Clavien It is not reproduced. Table information comes from the main text of the citated article (reference [21]) Dindo Classification of surgical complications. From Clavien et al. [23].

Degree	Definition
I	Any deviation from the normal postoperative course, without need for intervention beyond the administration of anti-emetics, antipyretic, analgesics, diuretics, electrolytes, and psychical therapy ^a^.
II	Complication requiring pharmacological treatment with other medicines beyond the ones used for complication of degree I.
III	Complications requiring surgical, endoscopic, or radiological intervention.
III-a	Intervention without general anaesthesia.
III-b	Intervention under general anaesthesia.
IV	Life-threatening complication requiring admission to intensive care unit.
IV-a	Uniorgan dysfunction (including dialysis).
IV-b	Multiorgan dysfunction.
V	Death.

^a^ This degree also includes wound infections opened at the bedside.

**Table 4 jpm-13-01540-t004:** Sociodemographic and clinical data of the study population (n = 51).

	n (%)
Age * (years)	59.86 (SD: 12.45)
Weight * (kg)	71.36 (SD: 16.13)
ASA PS classification	
*I*	2 (3.92%)
*II*	29 (56.86%)
*III*	20 (39.22%)
Tumour	
*Primary*	32 (62.75%)
*Recurrent*	18 (35.29%)
*Metastasis ***	
*Yes*	11 (22.45%)
*No*	38 (77.55%)
Neoadjuvant treatment **	
*Yes*	19 (38.78%)
*No*	30 (61.22%)

SD: standard deviation; ASA: American Society of Anesthesiologists; PS: physical status. * Values are presented in mean (standard deviation). ** Data were available for 49 patients.

**Table 5 jpm-13-01540-t005:** Preoperative, intraoperative, and postoperative data of study population (n = 51).

	Mean (SD)
Preoperative haemoglobin (g/dL)	11.19 (SD: 1.51)
Intraoperative PCI * [range: 0–39]	14.75 (SD: 9.25)
Blood loss (mL)	2.149.02 (SD:867.73)
Administered blood products (mL)	1.023.53 (SD: 637.37)
Fluid therapies [colloids + crystalloids] (mL)	8177.45 (SD: 3002.02)
*Colloids* (mL)	*887.25 (SD: 422.24)*
*Crystalloids* (mL)	*7290.2 (SD: 2879.95)*
Duration of surgery(min)	531.18 (SD: 121.4)
Postoperative lactic acid (mmol/L)	2.59 (SD: 1.83)

SD: standard deviation; PCI: peritoneal cancer index. * Data were not available for 3 patients.

**Table 6 jpm-13-01540-t006:** Correlation coefficients between lactic acid and other operative variables.

Variables	Correlation Coefficients (ρ)	*p*-Value
Preoperative haemoglobin	−0.144	0.313
Intraoperative PCI	0.532	<0.001 *
Blood loss	0.383	0.006 *
Administered blood products	0.326	0.021 *
Fluid therapies (colloids + crystalloids)	0.384	0.006 *
*Colloids*	0.298	0.036 *
*Crystalloids*	0.324	0.022 *
Duration of surgery	0.518	<0.001 *

PCI: peritoneal cancer index. * *p* < 0.05 (statistically significant result).

**Table 7 jpm-13-01540-t007:** Multivariate analysis. Linear regression model (lactic acid).

	Non-Standardized Coefficient		Standardized Coefficient	t	Sig.
	B	Standard Error	Beta		
Intercept	−1.058	1.079		−0.981	0.332
Duration of surgery (min)	0.005	0.002	0.336	2.330	0.024
PCI	0.060	0.029	0.297	2.054	0.046

PCI: peritoneal cancer index.

## Data Availability

Not applicable.

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
