# Peer review of "Analysis of Intraoperative Variables Responsible for the Increase in Lactic Acid in Patients Undergoing Debulking Surgery"

_jpm, 2023, doi:10.3390/jpm13111540_

Round 1
Reviewer 1 Report
Comments and Suggestions for Authors
Thank you for the opportunity to review this manuscript on relationship of cytoreductive surgery and lactic acid levels and implication of such.
This is a manuscript of clinical interest but requires improvement of presentation in the development of arguments in the introduction and discussion and reporting of methods and results.
Introduction - too many short paragraphs. Need to build an argument. Part of this argument needs to include why lactic acid is a measure of interest ie., what is known and what needs to be uncovered and why you did this study.
Line 47 - not sure what "abstentionist attitude" means. Need to be clear
Line 50,72 ect - author uses the same phrase "associated or not with". There could be better wording ie., with or without
Line 50-51 - not clear what "different cell lines" means.
Line 54 - give some examples of non-gyn neoplasms in bracketts.
Line 56 - From the context I think the authors means that PCI is the Sugarbaker's PCI in line 60. Need to make this clear.
Line 64 - is the PCI calculated before or after surgery - please be clear.
Methods:
List the included population. It is not clear if this is a study of ovarian cancer, uterine cancer, appendiceal cancer. I gather this is you 57 patients.
List the exclusion criteria. I assume this results in the 51 cases.
List the primary outcome - I assume this is the post op lactic acid level.
List secondary outcomes ie., HGB ect.
Results
Line 122 replace "have been retrospective" with the word "were".
Need to more clear say that your population includes x patients with primary debulking and 18 patients with debulking for recurrent disease.
Discussion
Line 212-213 is contradictory - your data shows higher HGB is asso to lower lactate levels - you data shows that this relationship is not significant (thus not important). You could say there was a non-statistically significant trend in the association of preop Hbg tp postop lactate levels
Line 227 - since the authors never tell the disease(s) that the patient had we can't actually tell if this is a homogenous cohort of patients
Line 239-243 Need to state what you found and what needs to happen next. ie., needs to be confirmed in other studies with a larger sample size, needs to be assessed using different intraoperative strategies in patient resuscitation, ect.
Comments on the Quality of English Language
Problem with verb tenses in some parts of the paper Line 91, 122,
Problem with some word usage ie., Line 47 abstentionist, line 46 incurable pathology
Problem with many paragraphs in introduction stating facts but without building a clear argument for why this study is needed.
Author Response
First, thank you very much for all your comments. I really appreciate them. It’s important to note that you do not have the last version of my manuscript. I think there has been a mistake because I was asked for an extended version of it and the one you have revised it is not that one. So, I have modified my last version with your corrections. I attach the new one. Hope you find it interesting and complete.
Please see the attachment.
Related to your comments:
INTRODUCTION
- The introduction has already been extended. I try to explain what peritoneal carcinomatosis is, the severity of the pathology, so it represents advanced malignant disease, and it was considered incurable some decades ago.
- Line 47. This is what abstentionist attitude means. Most of cases were lethal. The introduction of surgery in this kind of pathology, which strives to achieve complete cytoreduction, aims to increase survival and provide a definitive cure in some cases. Previously, patients had a very grim prognostic with just some months of survival, the treatment was palliative.
- Related to line 50-72, I do not have any problem in changing the words.
- Line 50-51. I added some examples of peritoneal carcinomatosis from different cell lines.
- Line 54. Examples added.
- Line 56. In my final manuscript version, classifications are thoroughly explained.
- Line 64: PCI calculation added.
METHODS
- Patients included were all patients operated from cytoreductive surgery between 2014 and 2016 regardless of the origin of carcinomatosis peritoneal (most of them had the origin in ovarian cancer, but there are also patients who had the origin in gastric, colorectal cancer...).
- The exclusion criteria, as it is said, it is the absence of information from the variables analyzed. It is a retrospective study and the variables analyzed were collected from patient medical records. Some of this information was missed out.
- The primary outcome of our manuscript is the linear bivariate correlation between lactic acid levels and intraoperative variables.
- The secondary outcomes are multivariate relation with lactic acid levels and explore linear correlation within intraoperative variables.
RESULTS
- Line 122 modified.
- From the 51 patients included in our study, 18 patients were recurrent tumours with carcinomatosis. Rest of the patients were primary tumours with carcinomatosis. In our results, this fact is not important. I could delete this differentiation but it is part of the sociodemographic data.
DISCUSSION
In my final manuscript version, I explain all these points. You can revise it and I can improve it if necessary.
Reviewer 2 Report
Comments and Suggestions for Authors
article mentions that increased hemoglobin is related to lower lactate levels, but this relationship was not statistically significant in the studied population. This suggests that the sample size may not have been large enough to detect a significant effect or the effect itself is not strong.
2. Limited Scope: The study focuses on specific intraoperative variables such as operative time, hemoglobin levels, and fluid therapy. However, other potential factors influencing lactic acid levels, such as patient's age, nutritional status, obesity, diabetes mellitus, tobacco use, coexistent remote body-site infections, altered immune response, corticosteroid therapy, recent surgical procedure, length of preoperative hospitalization, and colonization with microorganisms, are not discussed in the article.
3. Lack of Control Group: The study does not appear to include a control group for comparison, which could limit the strength of its conclusions. For example, it would be useful to compare the results with a group of patients who underwent similar surgeries but did not experience an increase in lactic acid levels.
4. Reliance on Empirical Values: The study mentions that fluid administration has been based on empirical values for a long time. This could be a limitation as it may not consider individual patient differences and could lead to over or underestimation of fluid needs.
5. Lack of Long-term Follow-up: The study does not seem to discuss the long-term effects or complications related to increased lactic acid in patients. Understanding the long-term implications could provide a more comprehensive view of the issue.
6. Potential for Bias: The study was conducted in a single institution, so the findings may not be generalizable to other settings. This could introduce bias, as a single institution's practices and patient populations may not reflect those in other hospitals or regions.
7. Lack of Multivariable Analysis: The study does not appear to use multivariable analysis to control for potential confounding factors. This could limit the accuracy of the findings, as it may not account for the influence of other variables on the increase of lactic acid levels.
8. Assumption of Linear Relationship: The study seems to assume a linear relationship between operative time and lactic acid levels, stating that the longer the intervention lasts, the higher the lactic levels of the patient. However, this may not always be the case, and a more complex relationship could exist. The study does not appear to explore this possibility.
9. Lack of Consideration for Preoperative Anemia: The article mentions that preoperative anemia is associated with worse outcomes after surgeries, with an increase in the number of transfusions that in turn, is associated with increased complications. However, it does not seem to consider the impact of preoperative anemia on lactic acid levels, which could be a significant oversight.
10. Assumption about Fluid Administration: The study assumes that greater fluid administration is related to longer surgical times, more complex surgeries, greater bleeding, and higher lactic levels. However, it does not appear to consider the possibility that the relationship could be bidirectional or influenced by other factors.
11. Lack of Consideration for Different Fluids: The study discusses using balanced crystalloids as the first fluid option. However, it does not seem to consider the potential impact of different types of fluids on lactic acid levels.
12. Assumption about Third Space: The study mentions that the concept of the third space has been abandoned, and goal-directed therapy, or zero-balance therapies, has been associated with better surgical outcomes. However, it does not provide evidence to support this claim, which could be a limitation.
Comments on the Quality of English Languagegood
Author Response
First, thank you very much for all your comments. I really appreciate them and find them interesting and motivators to future investigations. It’s important to note that you do not have the last version of my manuscript. I think there has been a mistake because I was asked for an extended version of it and the one you have revised it is not that one. So, I have modified my last version with your corrections. I attach the new one. Hope you find it interesting and complete.
Please see the attachment.
- In my last manuscript version, I have already commented about hemoglobin variable and the sample size.
-
This paper focuses on some intraoperative variables. There are a lot of preoperative and postoperative variables that can influence lactic acid levels and can be studied in subsequent investigations.
- Not all the patients experienced an important increase in lactic acid levels. That’s why I did this study. In the study we published in 2021, the relationship between lactacidemia and postoperative complications after CRS was analysed. It was observed that lactic acid levels on admission to the critical care unit were significantly higher in those who suffered complications, with a relative risk of almost threefold (2.9; CI95%: 1.6-5.3) with respect to uncomplicated patients, with 2.5 mmol/L being the cut-off point for an increased risk of complications and death. That means serum lactate level is a predictive factor for postoperative complications in patients undergoing CRS for peritoneal carcinomatosis. The study suggests that the risk of developing severe complications almost triples with a lactic acid level of 2.5 mmol/L or higher at the time of admission in the ICU. So, the search for factors that may contribute to the increase in blood lactacidemia levels is a major challenge in our work. That’s the reason to develop our paper and continue the investigation on this field.
- What it is said, it is that for a long time, fluid administration has been based on empirical values. Large amounts of crystalloids have been administered to replenish deficits assumed by fasting, blood loss, diuresis, insensible losses and the called third space. In systematic reviews that mediate extracellular volume, it was concluded that the third space was erroneous, and this concept has now been abandoned, and goal-directed therapy, or zero-balance therapies, have been associated with better surgical outcomes. In our institution the trend is to perform goal-guided therapy through minimally invasive monitoring (proAQT). ProAQT is a minimally invasive hemodynamic monitor which enables a reliable and physiological interpretation of the patient’s hemodynamic status. I had added this information in my last manuscript version. Even though, fluid therapy is a complex concept which should be individualised.
-
In my last manuscript version, I develop my previous publication on the topic, “Evaluation of the relationship between lactacidemia and postoperative complications after surgery for peritoneal carcinomatosis”. In this article, complications related to increased lactic acid in patients are developed.
- Potential for bias is true. That’s why I say the data come from a small series and a single centre, so the results should not be generalized.
- After bivariate analysis, a multivariate analysis was done with stepwise methodology to mitigate these possible confounding factors.
- In this kind of procedures, this fact uses to be the rule. These procedures are developed by the same multidisciplinary specialized team, so, the longer the interventions last mean more complex surgeries with a major volume blood loss and an increase in lactic acid.
- In my last manuscript version, it is outstanding that preoperative anaemia increases the need for perioperative blood transfusion, and there is now a well-recognized association with increased patient complications, length of hospital stays, and worse outcomes. In our study there is an association but this one is not statistically significant. Our data comes from a small sample of patients which may not have been large enough to detect a significant effect. Therefore, further studies with a larger population should be conducted to make final conclusions.
- This study does not imply causality. It shows bidirectional outcome measure such as correlation coefficient. To value the influence of other factors, a multivariate analysis with stepwise methodology has been applied.
- I do not understand exactly what you mean. In this study we have analyzed crystalloids and colloids separately. Even though, most of the fluid therapy used are crystalloids due to its known characteristics.
- In my last version, I add evidence about all these fluid therapies.